# Knowledge, Perceptions, and Perspectives of Medical Students Regarding the Use of Antibiotics and Antibiotic Resistance: A Qualitative Research in Galicia, Spain

**DOI:** 10.3390/antibiotics12030558

**Published:** 2023-03-11

**Authors:** Juan M. Vázquez-Lago, Rodrigo A. Montes-Villalba, Olalla Vázquez-Cancela, María Otero-Santiago, Ana López-Durán, Adolfo Figueiras

**Affiliations:** 1Service of Preventive Medicine and Public Health, Clinic Hospital of Santiago de Compostela, 15706 Santiago de Compostela, Spain; 2Health Research Institute of Santiago de Compostela (IDIS), 15706 Santiago de Compostela, Spain; 3Service of Admission and Clinical Documentation, Clinic Hospital of Santiago de Compostela, 15706 Santiago de Compostela, Spain; 4Department of Clinical Psychology and Psychobiology, University of Santiago de Compostela, 15782 Santiago de Compostela, Spain; 5Department of Preventive Medicine and Public Health, University of Santiago de Compostela, 15782 Santiago de Compostela, Spain; 6Consortium for Biomedical Research in Epidemiology & Public Health (CIBER en Epidemiología y Salud Pública—CIBERESP), 28029 Madrid, Spain

**Keywords:** antibiotic resistance, antibiotics, knowledge, perceptions, attitudes, medical students, qualitative

## Abstract

Antibiotic resistance is a significant public health concern, with numerous studies linking antibiotic consumption to the development of resistance. As medical students will play a pivotal role in prescribing antibiotics, this research aimed to identify their perceptions of current use and factors that could influence future inappropriate use of antibiotics. The study employed a qualitative research approach using Focus Group discussions (FGs) consisting of students from the final theoretical course of the Medicine degree. The FGs were conducted based on a pre-script developed from factors contributing to antibiotic misuse identified in previous studies. All sessions were recorded and transcribed for analysis by two independent researchers, with all participants signing informed consent. Seven focus groups were conducted, with a total of 35 participants. The study identified factors that could influence the future prescription of antibiotics, including the low applicability of knowledge, insecurity, clinical inertia, difficulties in the doctor-patient relationship, unawareness of available updates on the topic, and inability to assess their validity. The students did not perceive antibiotic resistance as a current problem. However, the study found several modifiable factors in medical students that could explain the misuse of antibiotics, and developing specific strategies could help improve their use.

## 1. Introduction

Antibiotic resistance is an increasingly significant public health issue worldwide [1,2], with substantial implications for morbidity, mortality, and costs [3,4]. There is now little doubt that the consumption of antibiotics is strongly linked to the development of resistance [3,5].

Spain’s antibiotic consumption is higher than the European Community average, despite no difference in infection prevalence [6,7]. This abuse and misuse of antibiotics is a complex issue that pertains to different groups, including doctors, healthcare users, pharmacists, veterinarians, and health authorities, and is related to knowledge, attitudes, and practices [3,8,9,10,11]. Medical students are an ideal population to implement educational strategies during their university studies to improve antibiotic prescription and use in the future. They are trained in the functionality of antibiotics, and their appropriate prescription [12,13] and are aware of the issue of antibiotic resistance. However, they may still lack confidence in selecting the right antibiotic for each case, providing instructions, and communicating with patients [12,13,14]. A systematic review of medical students’ knowledge, beliefs, and attitudes regarding antibiotics and resistance showed a need for more training to raise awareness of this public health problem [15].

Accordingly, our research aimed to explore the factors that influence the students of the last theoretical course of degree in Medicine on the future prescription of antibiotics and resistances in order to identify knowledge gaps on which to design future strategies.

## 2. Results

A total of 35 final-year medical students participated in seven focus groups, with each group consisting of 4 to 6 participants (see Table 1). 62.86% were women. None of the students invited to participate declined to take part in the study.

The initial group served as a pilot study, during which we introduced certain modifications to the script. Due to the qualitative methodology’s flexibility [16], we were able to incorporate new topics that emerged in this group into the subsequent sessions.

Through analysis of the transcripts, we identified students’ perceptions of current antibiotic use and the main factors that could lead medical professionals to abuse or misuse antibiotics in the future. Drawing on participants’ insights from the focus group, we compiled Table 2 to summarize the reasons behind this trend.

### 2.1. Knowledge about the Use of Antibiotics

All seven groups indicated that they were familiar with the general mechanism of action of antibiotics, the concept of “antibiotic resistance,” and the biological mechanisms of its development. However, a few students shared common misconceptions found among the general population concerning the symptoms that indicate the necessity for antibiotic treatment: “*[...] sputum if it is dense, green, then, antibiotic*” (F2, FG1); “*green mucus in children, earache, sore throat*” (M2, FG3), such claims were not refuted, nor discussed by their peers, who were sympathetic to them. They expressed their opinions about the mechanisms regarding the development of antibiotic resistance: “*if you always give fosfomycin, maybe that patient tends to generate a resistance to fosfomycin; if he has been taking it all his life, you also have to be careful with that*.” (F2, FG1).

### 2.2. Perception of Current Antibiotic Use

All seven groups concurred on the prevalence of antibiotic misuse, which they attributed to various factors, including both healthcare professionals and patients. The students linked the inappropriate use of antibiotics with certain behaviors they observed during medical consultations. For instance, they noted that inadequate patient histories and physical examinations often resulted in uncertain diagnoses, making it challenging to determine the appropriate treatment: “*[...] in the end what decides what is done is the time of anamnesis and exploration. It makes you more certain whether to give an antibiotic or not. If you have little time, you give things without evidence or don’t look for it so much*.” (M1, FG1).

They also pinpointed the insufficient explanations from doctors regarding the diagnosis, prognosis, treatment, and the significance of adhering to the prescribed dosage as another issue of antibiotic use. This inadequacy obstructs the doctor-patient relationship and results in noncompliance with the treatment plan “*[...] there are patients who, when they leave, have that desire of a better explanation of what they have and with a simple explanation they would better understand what they have, or they would be calmer*” (M2, FG6).

Two additional interconnected topics were identified: complacency in prescription and patient demand for antibiotics. Four groups referred to the issue of complacency in prescription “*[…] it is very tempting to give the antibiotic, and also the patient leaves with a smile. I know it shouldn’t be done, but I see people do it because of that: you watch your back and leave the patient happy*.” (M1, FG7). They said that doctors often succumb to this type of pressure on numerous occasions. Furthermore, it was noted that there is anticipation or expectation of patient demand “*if you think it will be driving you nuts for half an hour and you will end up giving it, then already ... you give it to him directly, don’t you*?” (M2, FG1).

In four groups, doctors’ lack of confidence in establishing the diagnosis and selecting treatment was identified as a contributing factor to prescription abuse. This lack of confidence was linked to the perceived specificity of symptoms, inadequate patient histories and physical examinations, and the absence of access to rapid tests, “*and I wonder: Doctor’s insecurity? The <I’m not sure, I’m not going to go out on a limb>, will that imply a lot? In theory, everything seems very easy, but a guy who is complaining there and you don’t know what he has…*”. Closely linked to the issue of insecurity, another concept emerged and was repeated in five groups: “defensive medicine.” This term refers to the practice of taking precautions to minimize the consequences for the doctor if their diagnostic hypothesis proves to be incorrect. The challenges of dealing with dissatisfied patients and providing additional assistance were also noted: “*in addition, you do not want to find him again in consultation if he returns and you did not give him an antibiotic ...*” (M1, FG5). This is also related to the reduction of risk for the patient. They reported that by prescribing antibiotics, they were providing coverage against potential complications of the patient’s condition: “*[...] many times, a pathology that has a very viral characteristic or seems very viral, to take care of health, they prescribe antibiotics. And that I think is one of the problems we have in the face of resistance: doctors are afraid of failing, or that a banal pathology gets complicated*” (F2, FG4), “*on the one hand, it is defensive of doctors, to save themselves in case this does not worsen, well*.” (F2, FG2).

The students mentioned that medical professionals might lack knowledge or experience circumstances that create doubt, ultimately leading to a prescription of antibiotics. This decision may be made in an effort to protect both themselves and the patient: “*what I saw in Primary Care was that many times they commented that there was a clear difference between the guidelines and clinical practice, and that was also manifested… there are some dogmas, and you block yourself to the theory*.” (M2, FG4).

Three groups identified inertia as a factor contributing to antibiotic misuse. However, it was noted that this was mentioned unconsciously in all three cases, as it was not subsequently linked to poor practices: “*[...] you give the best known by custom. So, you give this one out of habit because it goes well, it doesn’t have to be caused by the same bacteria, but you know it, it goes well, so I continue to use it*.” (F1, FG1). 

Five groups commented on the perception that the Pediatrics department is a service in which the use of antibiotics is commonly abused, primarily due to pressure or complacency from parents: “*Especially in Paediatrics, more than anything to calm parents, it’s like if you don’t give them antibiotics, you’re not doing your job properly*.” (F1, FG3).

### 2.3. Attribution of Responsibility for the Evolution of Antibiotic Resistance

Six groups assured that the current situation and the trajectory of antibiotic resistance development has a shared responsibility and a multifactorial cause “*is a bit everyone’s responsibility: what he says, patients taking them wrong, and doctors prescribing inappropriately, or because they ask for them... it’s something that affects everybody*” (F2, FG6).

The other group mentioned that once the healthcare sector became aware of the severity of the problem and began to regulate the use of antibiotics more strictly, the primary responsibility shifted to the general population. The public was perceived to be abusing antibiotics, not following treatment guidelines, and demanding them unnecessarily.

In three groups, the livestock sector was identified as a significant contributor to antibiotic resistance. Additionally, three students from different groups specifically highlighted the livestock sector as the primary source of antibiotic resistance “*partly the doctor, but also treatments to animals and the meat industry, which gives medicine and antibiotics to grow better, and I think it is also a fundamental part of the resistance*.” (F2, FG2).

### 2.4. Perception of the Magnitude of the Problem of Antibiotic Resistance

Although most participants acknowledged the severity of antibiotic resistance, their statements were not consistently aligned. Specifically, undergraduate Medicine students acknowledged the seriousness of the problem, recognizing that the advancement of resistance is outpacing the development of new antibiotics, which could result in a post-antibiotic era reminiscent of pre-antibiotic times: “*yes, we will return almost to the pre-antibiotic era*.” (F1, FG4). 

Regarding the prevalence of antibiotic resistance in our environment, participants stated that it is a medium to long-term problem. While they were aware of outbreaks of multidrug-resistant bacteria in the environment, they believed that these situations were specific and self-limited, part of a gradual progression, and that, in general, they had not observed them during their clinical practices. “*I, for example, think it is something rather global because I have never found a case in my training where I had to say <now this one does not work anymore>*.” (F1, FG4). 

### 2.5. Training

#### 2.5.1. Theoretical Training

The students acknowledged that their theoretical training was extensive and detailed. “*I think that they harped upon us a lot and informed us quite well*.” (M2, FG3). However, they also expressed that the knowledge they gained lacked practical application. “*adapting an antibiotic to a pathology is something I think that we were not taught at any time*.” (F4, FG2). As a result, we assessed their theoretical education as insufficient in terms of preparing them for professional practice: “*I believe that the training period is bad. I think that the training regarding antibiotics the form is not good, (…); I do not see the practical purpose of that way, and it is a problem during the whole degree, not only in the case of antibiotics*.” (M1, FG1).

Several students have expressed their perception of a significant disparity between the theoretical knowledge they have acquired and the practical skills exhibited by the doctors they have worked with. Additionally, these students have shared their concerns (fear) about feeling compelled to act in a manner that may contradict their own beliefs or values to adhere to hierarchical structures: “*we were in the health center in clinical sessions, and doctors always made a distinction between what is practice and theory in the prescription of antibiotics. I find it curious because they end up determining misuse because many of the theoretical criteria are not met in practice*” (M2, FG4).

#### 2.5.2. Tools and Skills

The students unanimously reflected that they lacked the essential skills required to translate their theoretical knowledge into practical applications and establish effective doctor-patient relationships. They also noted that they lacked social and communicative skills that are essential for dealing with the pressures that arise in clinical settings. The students expressed their belief that their practical training had been inadequate in these areas, leaving them ill-prepared for the challenges of real-world practice: “*in the end, it’s a job where you deal with public face to face, and these skills are facing the public. They don’t put a lot of emphasis on this in training*.” (M1, GF6), “*during the degree we are not taught any communicative skill. They teach you to study; you learn that, and... you would have to know and understand it very well to be able to change it and explain it with your words well. And we don’t do that during the degree*.” (F2, FG6).

#### 2.5.3. Update

According to the observations made during their practices, the most frequently utilized sources of information by the students for updates were conferences, clinical sessions, and clinical practice guidelines. These sources were considered to be both accessible and reliable, and the students expressed a willingness to continue using them in the future.

However, the students also reported being uninformed about other available resources and lacking the necessary training to effectively utilize them for ongoing learning and professional development. They further noted that their reliance on note-taking as a primary study method had led them to overlook alternative sources of information beyond textbooks: “*That is, it would be good to know those websites or databases that are more reliable, other than searching on Medline or Wikipedia*” (M1, FG6).

### 2.6. About the Doctor-Patient Relationship

Without exception, the students acknowledged the critical importance of establishing and maintaining a positive doctor-patient relationship as a means of managing stress and effectively communicating to patients that antibiotic treatment may not be necessary for their particular condition during consultations: “*if you have confidence in your doctor, you trust him blindly and if he says no, that means no*” (M2, FG3).

### 2.7. Solutions

When asked about potential solutions to reduce antibiotic abuse, students identified education of the population and packaging tailored to the treatment duration as the most effective measures. 

Table 3 demonstrates the saturation of information gathered on factors contributing to inappropriate future prescriptions by medical students. This section may be subdivided into headings to provide a clear and concise description of the experimental results, their interpretation, and the conclusions that can be drawn from the experiment.

## 3. Discussion

This is the first qualitative study in Spain to examine the factors that influence medical students in their attitudes toward antibiotic use and resistance. The findings indicate that students recognize their role in combating antibiotic resistance but are hindered by a lack of understanding of basic concepts, limited practical experience, insecurity, inertia, and challenges in the doctor-patient relationship. Identifying these factors can inform the development of targeted strategies to improve antibiotic use and enhance the impact of interventions aimed at addressing these deficiencies.

Some beliefs have been identified in a similar study on the general population [10]. This study conducted on the general population has identified certain beliefs that are incorrect because they are based on outdated knowledge and not supported by current scientific evidence. For instance, some people believe that the color of mucus in upper respiratory tract infections is correlated with its etiology. Given that medical students are a group that falls midway between the general population and medical professionals, it is logical that they may share some of these opinions. This finding is consistent with other studies that evaluated the knowledge of medical students regarding the effectiveness of antibiotics in treating colds, influenza, and coughs. Surprisingly, only 47–60% of students knew that antibiotics were not the preferred treatment option [17,18,19].

Furthermore, despite claiming to know about the appropriate use of antibiotics and antibiotic resistance, some students’ statements indicate clear ignorance of these topics. For instance, confusion between the terms antibiotic resistance and tolerance, as well as resistance, pan-resistance, and therapeutic failure, has been identified among certain students. Confusion between some of these terms has been perceived equally among the general population [10,11]. This lack of understanding is consistent with findings from a systematic review by Nogueira–Uzal et al. [15], published in 2020, which reported a general lack of knowledge regarding the diagnosis and treatment of infectious diseases, particularly upper respiratory tract infections, among medical students, regardless of their level of study.

Students are aware of the abuse and misuse of antibiotics and how it leads to increased antibiotic resistance, which they associate with both doctors and patients. Although they agree that prescribers bear direct responsibility, they only partially attribute it to clinicians and attribute bad practices to external causes, primarily the lack of time during consultations. Previous studies have also identified a lack of time as a crucial factor in antibiotic prescription [20]. This not only limits the doctor-patient relationship but also hinders proper anamnesis and exploration and instruction of patients. In general, students relate the shortage of personnel to these issues.

According to the students, insecurity among doctors is one of the main causes of antibiotic prescription abuse. Although they have received extensive theoretical training and possess the necessary knowledge to manage infectious diseases and antibiotics, they express insecurity when faced with the actual clinical environment. Previous studies on medical students have also identified similar insecurities regarding the selection and dosage of antibiotic drugs, attributed to a low transferability of knowledge to practical environments [12,13,21,22]. However, some studies suggest that overconfidence may also contribute to poor prescribing practices [13,22].

The students also expressed their opinion that antibiotic abuse is more prevalent in pediatric services due to the demands made by parents for prescriptions. However, a similar study on parents of primary school children in the same community observed that this group is more aware of the function of antibiotics and is more likely to conform to the explanations provided by the clinician, especially if it comes from their usual Pediatrician. Nonetheless, pediatricians acknowledge that parents often request antibiotics out of fear [11].

The lack of communication and social skills necessary to establish a good doctor-patient relationship and convince demanding patients of the unnecessary use of antibiotics is another contributing factor to prescription abuse. The students claim that they have not been trained in this area. Both the students and other studies have emphasized that the doctor-patient relationship is crucial for proper antibiotic prescription by professionals and their appropriate use by patients [9,20,23]. This lack of communication skills often leads to giving in to patient pressure and promotes complacency in prescribing, which has also been observed in pharmacists and primary care physicians [9,24,25].

Although the students are aware of the severity and consequences of the increase in antibiotic resistance, they view it as a medium-to-long-term problem. They are uncertain about the extent to which it currently affects their healthcare environment, a belief shared by medical professionals across different fields [25,26,27]. Similar findings were reported in a systematic review, which revealed that students acknowledge antibiotic resistance as a global public health concern but do not express concern about its impact in their immediate workplace or learning environment, such as their teaching hospital [15].

### Strengths and Limitations

The students who participated in the focus groups were recruited from a single university and may not necessarily represent all students from public universities in the country. Therefore, caution should be taken when generalizing the results to other regions or countries. Nevertheless, qualitative methods aim to capture a range of perspectives, and generalizability is not typically an expected attribute of this type of research.

The methodology and design used in this study met all the points of the Consolidated Criteria for Reporting Qualitative Research (COREQ) scale [28], indicating that it adhered to the quality criteria required for qualitative studies (Appendix A). 

Qualitative methodology is of great interest as a tool for exploring and identifying attitudes related to the use of antibiotics that cannot be identified “a priori” by epidemiological studies with quantitative methodology included in the literature review since people’s behavior is strongly influenced by the cultural characteristics of the population where they live and the interpersonal relationships that are generated. This methodology seeks to understand reality and phenomena from the perspective of the individuals who experience them [29,30,31].

Seven focus groups were conducted, which accounted for the number of enrolled students. This allowed for the collection of information from a diverse range of perspectives about students’ perceptions and perspectives on antibiotic use at the end of their university academic stage. A total of 35 medical students participated. The sampling method used was simple random sampling and convenience sampling. In qualitative research, is a greater interest in analyzing and delving into the study cases without any loss of scientific rigor. As explained by Hernández, Fernández, and Baptista: “*In qualitative studies, the sample size is not important from a probabilistic perspective because the researcher’s interest is not to generalize the results of their study to a wider population. What is sought in qualitative research is depth. We are concerned with cases (participants, people, organizations, events, animals, facts, etc.) that help us understand the phenomenon under study and answer research questions.* [32]”

In this context, the sample size is therefore determined by the ability of the different focus groups to generate the necessary information (data) for the study. Information collection through focus groups involves forming groups, each with between 4 and 10 participants until information saturation is reached. This means that all possible ideas that we explore have already emerged from group discourses or discussions and that no new ideas are emerging. Therefore, if we continue to form focus groups, they will no longer provide new data for the study. In our case, this occurred with seven groups, totaling 35 medical students. Sample size in this type of study is not fixed “a priori” based on statistical calculations but rather is determined “a posteriori” by reaching the sample size considered correct when no new information is generated [33,34].

## 4. Materials and Methods

### 4.1. Research Design

A qualitative study was conducted using FG discussions to collect narrative data from students in the final theoretical course of their Medicine degree. This approach allowed for an exploration of the beliefs and perceptions of the student population regarding the use and misuse of antibiotics. The objective was to obtain a comprehensive and detailed description of the student’s beliefs and perspectives and to develop a theory-based justification using systematically collected information. The use of FGs enabled an in-depth exploration of the topic and provided valuable insights into the participants’ perspectives.

### 4.2. Target Population

FGs were conducted at the University of Santiago de Compostela (USC), which is the only university in Galicia, a region in northwest Spain, that offers a degree in Medicine. From 2010-2017, the university offered an average of 353.75 places per year, making it the public faculty with the highest number of places offered for this degree in the country.

### 4.3. Selection, Sample and Procedure

The Medicine degree program is a six-year curriculum that includes internships in health centers, clinical settings, and surgical services during the final year. For this study, participants were recruited from the School of Medicine at USC and were in their fifth year of study. This year of study is focused on theoretical training and is common to all participants, which created a relaxed atmosphere and facilitated the open expression of opinions and beliefs. Additionally, participants’ similar age, levels of knowledge, and educational experiences allowed for the discussion of diverse perspectives without communication limitations [16].

To guide the focus group discussions, a script was developed by drawing on findings from previous studies involving family doctors [9,35], community pharmacists from Galicia and Portugal [9], the general population [10], and parents of primary school students [11]. The script aimed to explore the reasons that may lead students to misuse antibiotics toward the end of their medical training. Additionally, the script aimed to examine whether the improper use of antibiotics observed among primary care doctors could be attributed to inadequate training or the adoption of certain practices and misuse of resources once in practice.

The focus groups were conducted in A Coruña between February and May 2018 using a random selection procedure. The students were personally contacted during their face-to-face practicals on Preventive Medicine and Public Health subject and were invited to participate in the study by the researchers, who were independent of the teaching staff and faculty. The researchers explained the study’s objectives and the nature of their participation. It is worth noting that the students did not have any previous relationship with the researchers.

FGs sessions took place in a classroom located in the University Clinical Hospital of Santiago, which allowed for the participation of students who attended their theoretical classes in a building attached to the hospital. The room was occupied solely by the participants and the researchers, and contact information, such as email addresses, was collected from each group member. The focus groups were conducted by 2 resident physicians of Preventive Medicine and Public Health, 1 female (OVC) and 1 male (RAMV), both of whom had prior experience leading focus groups. One researcher acted as the interviewer, while the other served as a moderator to ensure that all group members participated in a respectful and organized manner. This approach fostered an environment conducive to the free expression of opinions and facilitated accurate transcription of the recordings.

The FG sessions were recorded using a digital recorder and a mobile phone to ensure high-quality sound for transcription purposes. The average duration of the sessions was 39 min, and they continued until no new ideas were presented. No data management software was utilized in the study.

One of the researchers transcribed the sessions, and another researcher checked for accuracy. The FGs were identified as “StudentsGF 1-7”, and each participant was assigned a code consisting of a letter “M” or “F” (to indicate male or female, respectively) and a number based on their speaking order in the audio files.

After each session, the two researchers discussed their initial impressions and noted down the group’s characteristics. Additional focus groups were formed until “saturation” was achieved, meaning that no new information was provided by the participants. At this point, adding more units would have been redundant and would not have improved the quality of the study [36].

### 4.4. Ethical Considerations

The study underwent evaluation and received approval from the Santiago-Lugo Research Ethics Committee, registered under code 2014/386. Prior to participation, participants were informed of the study’s objectives and the intention to record and transcribe sessions. They provided their consent to participate by signing an informed consent form. The study ensures the anonymity of all participants.

### 4.5. Analysis

The analysis of the transcripts was a repetitive process that involved two independent researchers. They were responsible for carefully reading the transcripts to ensure an appropriate structure of the data, which allowed for a deeper interpretation and reduced the risk of researcher bias.

Thematic and discursive content analysis was employed to examine the data, enabling the identification of different ideas and organization of the obtained data into relevant topics, supported by literal extracts serving as units of analysis [37]. The extracted ideas were then associated with the pre-established variables. In cases where there were disagreements between the researchers regarding the interpretation, they were debated and resolved by consensus. Given the limited number of focus groups, no software was used for data processing. The definition of the FGs was based on the participants’ use of different concepts (Table 4).

## 5. Conclusions

This study highlights that Undergraduate medicine students lack adequate theoretical training in the prescription and use of antibiotics. Furthermore, they do not find the little training they receive clinically applicable. Additionally, they attribute their method of prescribing antibiotics to inertia, copying other professionals. 

This study is a first step, which will allow the design of a validated questionnaire from which multifaceted interventions and strategies can be designed to improve the prescription of future medical professionals.

## Figures and Tables

**Table 1 antibiotics-12-00558-t001:** Focal Groups characteristics.

	n	M-F
**FG1**	5	2-3
**FG2**	5	0-5
**FG3**	5	2-3
**FG4**	4	2-2
**FG5**	6	4-2
**FG6**	5	2-3
**FG7**	5	1-4

FG. Focal Groups; M: Male; F: Female.

**Table 2 antibiotics-12-00558-t002:** Factors identified with respect to knowledge and attitudes regarding antibiotic use among medical students.

Knowledge		They claim to know the concept of antibiotic resistance and the mechanisms by which they are developed.
Perception	Use	They perceive abuse of antibiotics. They perceive pressure for the prescription to which it is yielded. They perceive inertia on the part of professionals.
	Responsibility	Multifactorial: ○ Doctors. ○ Patients: adherence, leftover kits. ○ Food sector.
	Magnitude	Aware of the seriousness of the advance of the resistance. Unaware of the degree of presence of resistance in their healthcare environment.
Training	Theoretical	They claim to have received good theoretical training. They believe that their knowledge has no practical applicability.
	Skills and tools	They report poor practical training: they lack skills and assertiveness. Insecurity before the diagnosis.
	Updating	They perceive little updating among doctors. Through congresses, clinical sessions, and clinical guidelines. Lack of information search tools and critical reading.
Perspectives	Doctor-patient relationship	They consider it essential. They report lacking the necessary skills and time.
	Training Industry	Awareness of the existence of biases. Perceived as necessary.

**Table 3 antibiotics-12-00558-t003:** Saturation of information on identified factors contributing to inadequate future prescribing.

Contributing Factors to Future Bad Prescribing	FG1	FG2	FG3	FG4	FG5	FG6	FG7
Low practical applicability of knowledge	X	X	-	X	X	X	X
Lack of social and communication skills	X	X	X	X	X	X	X
Lack of knowledge of updating tools and continuous training	-	X	X	X	X	X	X
Need for the industry as a trainer	-	X	X	X	X	X	X
Insecurity	X	X	X	X	X	X	X
Clinical inertia as a valid tool	X	X	-	X	-	X	X
Patient demands	X	X	X	X	X	X	X
Lack of awareness of the current presence of antibiotic resistance in the direct environment	X	-	X	X	-	X	X

**Table 4 antibiotics-12-00558-t004:** Concept coding.

Concept	Definition According to Its Use
Update	Methods they know or observe to keep knowledge up to date.
Complacency	Unnecessary prescription for meeting the expectations perceived in the patient.
Skills	Social and communicative skills available to establish a good doctor-patient relationship. Ability to set limits and not give in to patient demands.
Tools	Ability to bring their theoretical knowledge to the practical field. Means available to them to solve doubts individually.
Training Industry	Assessment and perception of the pharmaceutical industry as a method of updating and continuous training.
Inertia	Tendency to use the same treatments in similar situations without inquiring into the indication because: - had worked in the past in other cases. - a colleague would advise or order it. - is the usual treatment used in the service.
Magnitude	- Severity and extent perceived on antibiotic resistance.
Defensive Medicine	Proceeding perceived as: - less risky in possible repercussions for the professional. - of lower risk for the patient by covering possible complications.
Perception	What students claim to observe in clinical practice.
Pressure	User demand to be prescribed an antibiotic.
Responsibility	Attributed guilt to the development of antibiotic resistance.

## Data Availability

The transcriptions used and/or analyzed during the current study are available from the corresponding author upon reasonable request.

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
