# Peer review of "Knowledge, Perceptions, and Perspectives of Medical Students Regarding the Use of Antibiotics and Antibiotic Resistance: A Qualitative Research in Galicia, Spain"

_antibiotics, 2023, doi:10.3390/antibiotics12030558_

Round 1
Reviewer 1 Report
I have read with interest this manuscript by Vazquez-Lago et al., where the authors report the methodology and results of a qualitative research on the knowledge, perceptions, and perspectives of medical students regarding the use of antibiotics and antibiotic resistance in a university in the region of Galicia, Spain.
The study is interesting and its results add one more piece of evidence about the knowledge, perceptions and perspectives, which would help to target better-focused training actions for medical students to improve the appropriate use of antimicrobials once they complete their degree.
I have some comments:
Major comment:
The title may mislead the reader about the scope of the study. This is neither a multicenter nor a multinational study. Therefore, regarding this kind of qualitative research with a limited sample of individuals from a single centre, it is of paramount importance that the title should clearly include the geographical scope of the study. There are numerous articles reporting similar qualitative studies performed in different locations (e. g. the output of a quick search in PubMed resulting from the search term [knowledge+undergraduate+antibiotic] returns 130 manuscripts, most of them indicating in their title the geographical location where the study was conducted). Adding only "Spain" to the title would not address this issue, as the students who participated in the study were recruited from the only university that offers a degree in Medicine in Galicia as the authors point out, thus it cannot be considered as representative of the medical students in the whole country. I strongly suggest rewording it as "...medical students in Galicia, Spain", which would undoubtedly inform the reader in the title where and to what extent the study was performed.
Minor comments:
Affiliations: Please justify the affiliations to the left.
Introduction section:
-. Lines 49-50: The two references used by the authors to support the statement about the ranking of Spain in antimicrobial consumption with respect to the average figure in Europe are quite out of date. I strongly suggest presenting more up-to-date references (i.e. from ECDC latest ESAC report: https://www.ecdc.europa.eu/sites/default/files/documents/ESAC-Net_AER_2021_final-rev.pdf), instead of only manuscripts presenting data from before 2010.
Results section:
-. Line 115: Please use the names of antibiotics with a lower-case first letter ("fosfomycin" instead of "Fosfomycin") unless it is the first word after a stop. (https://community.cochrane.org/style-manual/grammar-punctuation-and-writing-style/upper-case-letters#:~:text=Pharmaceutical%20drug%20brand%20names%2C%20if,names%20should%20not%20be%20capitalized.)
-. Line 138: Please replace the acronym "AB" with "antibiotic", as that acronym is neither defined previously nor used in the rest of the manuscript.
-. Line 166: Please do not use the acronym "AP". Instead, use the developed definition. Is it "antibiotic prescription" or "antibiotic prescribing"? or even is "Atencion Primaria" in Spanish, that is "Primary Care" in English? Please clarify.
-. Lines 178-179: The sentence between quotes: “Especially in paediatrics, more than anything to calm parents, it's like if you don't give them antibiotics you're not doing your job properly." should be in italics as those other sentences of the same kind in the manuscript. And use Paedriatics with the first capital letter.
-. Line 183: Which groups are "The groups assured ...". It seems that a number is missing in that sentence. Please revise to show how many groups in this case, as the next paragraph in the text refers to "The other groups".
Discussion section:
-. Line 298: It is not clear to the reader why the authors indicate that the study ref.10 responds to obsolete evidence, when that study was published in 2021. I suggest rewording or clarifying in the text the meaning of that sentence.
-. Line 312: The sentence "The confusion between some of these terms has been perceived equally among the general population" does not seem to be here in the right place, as the preceding paragraph does not discuss about terms. Please rephrase or locate it in the right place in the manuscript.
-. Line 335: The word "pediatric" is spelt here in American English, while in other parts of the manuscript is spelt in British English as "paediatric". Please, maintain consistency in language all over the text. Same comment about "pediatrician" in line 339.
-. Lines 339-340: The meaning of the sentence: "However, they admit to demanding antibiotic drugs out of fear" is not clear. If the authors wanted to say that "paediatricians admit that parents demand antibiotics from them out of fear", please reword adequately.
Methods section:
-. Line 412: please consider using "practicals" instead of "practices".
Conclusions section:
-. Line 486: I suggest adding the validated questionnaire to the supplementary material in order to give the opportunity to the scientific community to use it to design specific interventions and strategies to improve the prescription of future medical professionals.
Supplementary materials section:
-. Lines 490-491: There is no information about the supplemental table (the COREQ checklist) in these lines.
Author Contributions section:
-. Lines 492 and 493 should be removed.
Data availability statement section:
-. Line 512: There is a typo at the beginning of the sentence: "he", instead of "The".
Reviewer 2 Report
Lago et al in their manuscript described the Knowledge, perceptions, and perspectives of medical students regarding the use of antibiotics and antibiotic resistance. The reviewer has following concern:
1. Introduction and discussion parts need to be improved
2. Line number 172 information is missing
3. Respondents number is small. Please provide the scientific evidence/information in what basis you selected the respondents number.
4. Is there any correlation among the AMU and AMR? It should also be evaluated.
5. Does knowledge influences respondents’ practices? Please provide it
6. English language needs to be improved
7. Conclusion is not up to the marks
Reviewer 3 Report
Following are the observations.
1) Introduction need more focus on title rather than to discuss unnesseasry stuff.
2) Statement of problem, hypothesis and impact of study is missing.
3) Results are not sofficient and without any statiistical rrelation. So advised to add.
4) Method is inadequate so it needs more details.
5) Results tables are not sufficient.
Reviewer 4 Report
The manuscript tackles an interesting topic of medicine students’ knowledge and attitudes towards antibiotics and antimicrobial resistance. It identified key factors that could contribute to their future prescribing practice and how those could be improved with educational interventions.
Introduction:
The content of the Introduction is mostly well-though-out, covered the main background points and led up to the aim of the study. However, it takes a lot of time to actually understand what was written. The Introduction needs an extensive rewrite to improve the English language and style. I would suggest proofreading by a native speaker or to use professional language editing services because it is difficult to comprehend.
Results, Discussion and Methods section (including the tables) suffer from the same issue of the poor use of English language. It was very hard to understand parts of those sections. Abstract needs rewriting as well.
Methods section:
There is no calculation of the necessary sample size, nor the justification for sample size. Please state how the sample size was calculated.
Also, please state how many students are enrolled in the fifth academic year, in the year when the FGs were conducted.
Authors stated: “FGs were developed in A Coruña from February to May 2018” Does that mean they were conducted during that time period, or were being developed? If they were developed in that time, please state the time frame when they were conducted.
The content of the results and discussion section seems to be fine, at least those parts that are comprehensible. The results were mostly adequately interpreted and given the broader context in the discussion section.
Some of the references are outdated and the authors should find newer sources.
Overall, this is an interesting study that could be of the interest to the readers. However, the manuscript is written in a poor English and needs a serious rewrite to improve the language and style. Also, there are concerns with validity of the study as justification for the sample size was not provided.
Round 2
Reviewer 1 Report
Dear authors,
Thank you for addressing the comments in this second revision of your manuscript. I find it now improved and all the comments have been adequately addressed. In my opinion, the paper is now acceptable in its present form, although some -really minor- issues should be finally addressed in the clean final version, with no further revision on my side needed:
-. Affiliations 1 and 2 still are not justified to the left.
-. Delete the remaining "AB" in the second paragraph on page 33, as well as the remaining "AP" in the fourth paragraph on the same page.
-. Remove the yellow highlights in the text on page 41.
I would like to thank the authors for making the revision process so smooth and scientifically sound with the result of an improved manuscript as it is now.
Reviewer 2 Report
As many important issues raised by the reviewer, the author solved it by telling that this study is a qualitative study which lower the impact of the manuscript.
Some parts/information of The author response should be included in the discussion section.
Reviewer 3 Report
Paper revised according to the comments.
Reviewer 4 Report
Dear authors,
thank you, all my comments have been adequately addresed.
The manuscript is sufficiently improved and suitable for publishing.
